# Refining multi-model projections of temperature extremes by evaluation against land-atmosphere coupling diagnostics

Sebastian Sippel[1,2], Jakob Zscheischler[2], Miguel D. Mahecha[1], Rene Orth[2], Markus Reichstein[1], Martha Vogel[2], and Sonia I. Seneviratne[2]

[1]Max Planck Institute for Biogeochemistry, Hans-Knöll-Str. 10, 07745 Jena, Germany.
[2]Institute for Atmospheric and Climate Science, ETH Zürich, 8075 Zürich, Switzerland.

*Correspondence to:* Sebastian Sippel (ssippel@bgc-jena.mpg.de)

**Abstract.** The Earth's land surface and the atmosphere are strongly interlinked through the exchange of energy and matter. This coupled behaviour causes various land-atmosphere feedbacks, and an insufficient understanding of these feedbacks contributes to uncertain global climate model projections. For example, a crucial role of the land surface in exacerbating summer heat waves in mid-latitude regions has been identified empirically for high-impact heat waves, but individual climate models differ
widely in their respective representation of land-atmosphere coupling. Here, we compile an ensemble of 54 combinations of observations-based temperature (T) and evapotranspiration (ET) benchmarking datasets and investigate coincidences of T anomalies with ET anomalies as a proxy for land-atmosphere interactions during periods of anomalously warm temperatures. First, we demontrate that a large fraction of state-of-the-art climate models from the Coupled Model Intercomparison Project (CMIP5) archive produces systematically too frequent coincidences of high T anomalies with negative ET anomalies in mid-
latitude regions during the warm season and in several tropical regions year-round. These coincidences (high T, low ET) are closely related to the representation of temperature variability and extremes across the multi-model ensemble. Second, we derive a land-coupling constraint based on the spread of the T-ET datasets and consequently retain only a subset of CMIP5 models that produce a land-coupling behaviour that is compatible with these benchmark estimates. The constrained multi-model simulations exhibit more realistic temperature extremes of reduced magnitude in present climate in regions where
models show substantial spread in T-ET coupling, i.e. biases in the model ensemble are consistently reduced. Also the multi-model simulations for the coming decades display decreased absolute temperature extremes in the constrained ensemble. On the other hand, the differences between projected and present-day climate extremes are affected to a lesser extent by the applied constraint, i.e. projected changes are reduced locally by around 0.5°C to 1°C - but this remains a local effect in regions that are highly sensitive to land-atmosphere coupling. In summary, our approach offers a physically consistent, diagnostic-based avenue
to evaluate multi-model ensembles, and subsequently reduce model biases in simulated and projected extreme temperatures.

## 1 Introduction

The exchange of matter and energy between the land surface and the atmosphere is a crucial feature of the Earth's climate (Seneviratne et al., 2010b; Bonan, 2015; van den Hurk et al., 2016). On one hand, the atmosphere exerts a key influence

on land surface processes such as vegetation growth by supplying light, water and carbon dioxide (Köppen, 1900). On the other hand, the land surface feeds back to the atmosphere, for example through the partitioning of energy into latent and sensible heat fluxes, or by modifying land surface properties, thus implying a direct link to near-surface climate (Koster et al., 2004; Seneviratne et al., 2010b). Conceptually, coupling between the atmosphere and the land surface is often classified into two qualitatively different regimes, a so-called "energy-limited" and "water-limited" regime (Seneviratne et al., 2010b): In the wet (energy-limited) regime, the land surface is largely controlled by the atmosphere through radiation (see conceptual Fig. 1a,b), implying a positive association between near-surface temperature (T) and evapotranspiration (ET). In contrast, in a dry, water-limited state, the land controls near-surface climate through a lack of soil moisture, and a corresponding reduction in evapotranspiration and latent cooling (see conceptual Fig. 1a,b) with a negative association between T and ET. Therefore, the state of the land surface and land-atmosphere feedbacks modulate and amplify climatic extreme events such as heat waves in mid-latitude regions (Seneviratne et al., 2006; Fischer et al., 2007; Hirschi et al., 2011; Whan et al., 2015; Hauser et al., 2016). An understanding of these feedbacks might yield improved seasonal predictability of extremes (Quesada et al., 2012), and could help to constrain and better predict model-simulated present and future climate variability in these regions (Seneviratne et al., 2006; Lorenz et al., 2012; Dirmeyer et al., 2013; Seneviratne et al., 2013; van den Hurk et al., 2016; Davin et al., 2016).

However, at present large uncertainties and methodological inconsistencies prevail in both understanding and quantification of land-atmosphere coupling at various spatial and temporal scales, which relate to

i. scarcity of accurate observational products of soil moisture or evapotranspiration at large spatiotemporal scales and relatively short observational periods (Seneviratne et al., 2010b),

ii. the metrics and variables used to quantify land-atmosphere coupling differ widely in the variables they address (Seneviratne et al., 2010b), and in emphasizing either the whole distribution (Dirmeyer, 2011; Lorenz et al., 2012; Miralles et al., 2012), or the tails of relevant variables (Zscheischler et al., 2015).

As a consequence, uncertainties and methodological inconsistencies contribute to a greatly diverging representation of land-atmosphere coupling in state-of-the art climate models (Koster et al., 2004; Boé and Terray, 2008, see also Fig. 1a,b for a simple conceptual example), and further contribute to uncertainties related to projected increases in summer temperature variability in the 21st century in mid-latitude regions (Seneviratne et al., 2006; Dirmeyer et al., 2013). In this context, it has been noted that accurate simulations of temperature variability and extremes require a realistic representation of land-atmosphere interactions (Seneviratne et al., 2006; Fischer et al., 2012; Bellprat et al., 2013). In other words, biases in temperature variability and extremes might in part stem from an unrealistic representation of land-atmosphere interactions (Fischer et al., 2012; Lorenz et al., 2012; Davin et al., 2016), likely leading to temperature-dependent biases in multi-model ensembles (Boberg and Christensen, 2012; Bellprat et al., 2013).

A model evaluation focus on interpretable land-atmosphere coupling diagnostics might serve as a complementary strategy to traditional model validation and testing (Seneviratne et al., 2010a; Santanello et al., 2010; Mueller et al., 2011b; Mueller and Seneviratne, 2014). Hence, this approach is intended towards testing and understanding the spread and physical consistency in simulated relationships in state-of-the-art multi-model ensembles (e.g. the Coupled Model Intercomparison Project, CMIP5

Taylor et al., 2012) against available observations-based datasets. For example, in the context of land-atmosphere coupling, earlier studies used bivariate correlation- or regression-based metrics to test and evaluate coupling behaviour (Hirschi et al., 2011; Lorenz et al., 2012). Conceptually, the notion of "diagnostic-based model evaluation" as discussed here is consistent with so-called "pattern-oriented model evaluation" (Grimm and Railsback, 2012; Reichstein et al., 2011) - the latter being

applied in the context of evaluating simulated and observed patterns at multiple scales in a data-driven way (e.g. in the context of ecosystem carbon turnover times, Carvalhais et al., 2014).

In the context of extracting credible and relevant information from large (multi-)model ensembles, weighting or selecting models based on observations-based constraints has become increasingly popular recently (Tebaldi and Knutti, 2007; Knutti, 2010), as a priori model ensembles might be seen as a somewhat arbitrary collection of model runs (or "ensembles of oppor-

tunity"). For example, empirical and/or physics-based criteria have been used to constrain snow-albedo feedbacks (Hall and Qu, 2006), constrain carbon cycle projections (Cox et al., 2013; Wenzel et al., 2014; Mystakidis et al., 2016), or in the context of refining precipitation projections (Orth et al., 2016). Moreover, empirical diagnostics are applied to select models for event attribution analyses (Perkins et al., 2007; King et al., 2016; Otto et al., 2015) and analyses of drought projections based on model performance (Van Huijgevoort et al., 2014), or to resample large initial-condition ensembles to alleviate biases without

distorting the multivariate structure of climate model output (Sippel et al., 2016b). In the context of land-atmosphere coupling, Fischer et al. (2012) and Stegehuis et al. (2013) have constrained a regional model ensemble over Europe using present-day interannual variability of summer temperature, and observations-based estimates of summer sensible heat fluxes. However, these studies came to somewhat conflicting results with respect to the obtained change in warming projections, which probably was due to the underlying choices of datasets to obtain the constraints (Stegehuis et al., 2013). Hence, care is needed in that these

practices might not necessarily translate into improved future climate projections or reduced uncertainties. That is because the selection of relevant metrics is clearly not trivial but subjective, and because good model performance w.r.t. any given metric does not translate directly into (more) reliable projections (Knutti, 2008).

Therefore, the starting point for the present analysis -in the sense of being necessary, but not sufficient to assure reliability of future climate projections- is that physically motivated, observations-based diagnostics might offer

1. a link to identify and interpret relevant processes across multiple models (i.e., model evaluation), and

2. to reduce biases by focusing the interpretation of multi-model ensembles on models that are "right for the right reasons". Most notably climate impacts, including extremes, typically depend on the multivariate structure of climate variables, where simple univariate statistical bias correction methods are prone to failure (Ehret et al., 2012; Cannon, 2016).

In this study, we first evaluate land-atmosphere coupling in state-of-the-art global climate models from the CMIP5 archive

and a large ensemble of observations-based ET datasets (Mueller et al., 2013) that has been compiled to address the aforementioned uncertainties in land-atmosphere coupling. In our analyses a land-atmosphere coupling metric that is based on coincidences of temperature and evapotranspiration anomalies is applied. The idea behind a coincidence metric as opposed to a traditional univariate evaluation of model simulated ET fluxes or temperature is that it is insensitive to biases in the simulated means or variances, and thus focusses only on an abstract property of the data, namely the bivariate dependence structure of T

and ET. Secondly, we derive a model constraint based on the physically motivated land-coupling diagnostic and the ensemble of benchmarking datasets in order to explore the implications of a reduced ensemble but with land-atmosphere coupling that is *within the range* of the benchmarking datasets.

## 2   Data & Methods

### 2.1   Datasets for T-ET coupling analysis and model evaluation

**Global temperature and evapotranspiration datasets**

In order to evaluate T-ET coupling in global climate models, an ensemble of 18 gridded ET estimates, taken from the
LandFlux-EVAL multi-data set synthesis project (Mueller et al., 2013), are combined with three different observations-based
and reanalysis-driven temperature datasets, yielding in total 54 T-ET combinations (see Table 1). T-ET coincidence rates are
calculated from each of those 54 combinations to evaluate and constrain the multi-model ensemble of global climate models
(Section 3). The ensemble of ET reference datasets has been generated by combining a wide range of different ET estimates,
consisting of five diagnostic (based on remote sensing or in-situ observations) products, five land surface models driven by
observed climate forcing and four reanalysis products (Mueller et al., 2013). The three temperature datasets are based on one
observational product (Climate Research Unit dataset, Harris et al., 2014) and two reanalysis products (ERA-Interim reanal-
ysis (ERAI, Dee et al., 2011), and Climate Forecast System Reanalysis (CFSR, Saha et al., 2010), see Table 1 for details).
The large number of T-ET dataset combinations is used in order to take uncertainties in both T- and ET datasets into account.
We have tested that the spread between individual ET datasets is substantially larger than the spread between individual T
datasets (not shown). This indicates that the largest source of uncertainty stems from the choice of ET dataset, and therefore we
consider only three different T datasets. Each of the 54 T-ET dataset combinations (denoted as "T-ET coupling benchmarks" in
the remainder of the paper) is consistently derived from observations, and thus can be expected to represent relevant features
in T-ET coupling under different assumptions that underlie diagnostic datasets, reanalyses and land surface models. Therefore,
these datasets represent a very large spread of plausible T-ET coupling estimates, and the spread can be considered as a conser-
vative benchmark for model evaluation (including observational noise, i.e. allowing a wide range of T-ET coupling in models).
However, it should be emphasized that the datasets are not independent realizations. Thus, we only use the spread of the T-ET
coupling benchmarks, but we do not interpret the probability distribution of dataset combinations.

For the analysis of historical and future simulations of the monthly maximum value of daily maximum temperatures (TXx)
in Section 3.2 we use ERA-Interim (Dee et al., 2011) as a reference dataset.

**Multi-model ensemble simulations**

The Climate Model Intercomparison Project (CMIP5) has been designed to allow for multi-model comparison and evaluation
studies (Taylor et al., 2012). Although large model spread, biases and uncertainties remain in the ensemble projections (Knutti
and Sedláček, 2013), for example with respect to extremes (Sillmann et al., 2013a), the water (Mueller et al., 2011b; Mueller
and Seneviratne, 2014), and land carbon cycle (Anav et al., 2013), the archive of standardized scenario-driven model experi-
ments provides one of the main avenues to study climate variability and change (e.g. (Stocker et al., 2013)), including present
and future climate extremes (Sillmann et al., 2013b; Seneviratne et al., 2016). We use one ensemble member from 37 individual
models or model variants (Table S1) to avoid unequal sample sizes in the multi-model ensembles. Furthermore, this choice is

made to assess variability in land-atmosphere coupling *across* models, because individual ensemble members from the same model show comparably small spread in land-atmosphere coupling and present-day and future land-atmosphere coupling are highly correlated (Fig. S1, metric and definition is provided below). This indicates that the large spread between models is dominated by variability *across* models, and thus land-atmosphere coupling is a model-inherent feature on climatological time

scales (Fig. S1 and Fig. S2, see further discussion below). On shorter (e.g. annual or seasonal) time scales, models indeed show substantial variability in their land-atmosphere coupling (Sippel et al., 2016b), which could be used as a constraint in large single-model ensembles but is beyond the scope of the present study.

**Data processing and analysis**

All datasets were remapped to a common $2.5° \times 2.5°$ spatial resolution for analysis and before computing T-ET coincidences.

For model evaluation (Section 3.1), all computations and analyses are performed on a monthly temporal resolution and are restricted to the time period 1989-2005 due to data availability constraints of the ET reference datasets (Mueller et al., 2013). Thus, the reference period for model evaluation corresponds to the last 17 years of the "historical" scenario in CMIP5 models. T-ET coincidences are computed based on monthly deseasonalized and linearly detrended time series of T and ET, and coincidence rates are calculated separately for each individual season. Only land pixels outside of desert regions following

the Köppen-Geiger climate classification are considered (Kottek et al., 2006). The model evaluation is conducted based on all individual pixels, and additionally on area-averages for so-called IPCC-SREX regions (IPCC, 2012).

## 2.2   Diagnostic-based model evaluation using T-ET coupling

**The T-ET link and the Vegetation-Atmosphere Coupling (VAC) Index**

An adequate characterization of the coupling between soil moisture and temperature is key to model evaluation using observations-

based datasets. This coupling is often diagnosed by correlation-based metrics such as for example between T and ET, $\rho_{(T,ET)}$ (Seneviratne et al., 2006; Lorenz et al., 2012), or the difference in the covariability of temperature and sensible heat, where the latter is calculated with and without accounting for soil moisture deficits (Miralles et al., 2012). Here, we aim to exploit the T-ET coupling by using a natural extension of $\rho_{(T,ET)}$ that focusses on the tails of T-ET dependedencies. Deseasonalized and detrended time series of ET ($x_i^{ET}$) and T ($x_i^T$, $i$ denotes the time step), are partitioned into five distinct classes of Vegetation-

Atmosphere Coupling (VAC) following (Zscheischler et al., 2015), resulting in a time series of discrete events $x_i^{VAC}$:

$$x_i^{VAC} = \begin{cases} a, & \text{if} & x_i^T < th_{lower}^T & \text{and} & x_i^{ET} < th_{lower}^{ET}, \\ b, & \text{if} & x_i^T > th_{upper}^T & \text{and} & x_i^{ET} > th_{upper}^{ET}, \\ c, & \text{if} & x_i^T > th_{upper}^T & \text{and} & x_i^{ET} < th_{lower}^{ET}, \\ d, & \text{if} & x_i^T < th_{lower}^T & \text{and} & x_i^{ET} > th_{upper}^{ET}, \\ 0 & \text{otherwise.} \end{cases}$$

Event thresholds $th_{lower}$ and $th_{upper}$ might be chosen relative to the variability of each time series by fixing the probability $p$ to exceed or fall below a threshold through the choice of an appropriate quantile:

$$Pr[X > th_{upper}] = Pr[X < th_{lower}] = p \tag{1}$$

Taking time series length restrictions into account, we choose the 30th and 70th percentile as lower and upper thresholds in all
time series (i.e. such that $Pr[X < th_{lower}] = Pr[X > th_{upper}] = 0.3$). Here, we focus on coincidences of *warm temperature anomalies* ("T-events": $x_i^T > th_{upper}^T$) with anomalies in ET ("ET-events", i.e. either $x_i^{ET} > th_{upper}^{ET}$ for $VAC_b$ or $x_i^{ET} < th_{lower}^{ET}$ for $VAC_c$), we derive coincidence rates $r_{VAC_b}$ by counting the number of $VAC_b$-events (see Quiroga et al. (2002); Donges et al. (2016) for earlier formulations of event coincidence analysis, and e.g. Rammig et al. (2014); Siegmund et al. (2016) for applications in an ecological context):

$$r_{VAC_b} = \frac{1}{N_0} \sum_{i=1}^{N} 1_{[b]}(x_i^{VAC})$$

Here, $1_A(x)$ is the indicator function, defined as $1_A(x) = 1$ if $x \epsilon A$ and $1_A(x) = 0$ otherwise, $N$ denotes the length of the time series. Hence, we simply count coincidences of T and ET in a given category (e.g. positive T *and* positive ET for $VAC_b$) to get the average coincidence rate ($r_{VAC_b}$). $N_0$ acts as a normalization constant and is chosen in our study such that $0 \leq r_{VAC_b} \leq 1$, i.e. we normalize with the total number of "T-events", $N_0 = \sum_{i=1}^{N} 1_{[x^T > th_{upper}^T]}(x_i^T)$. Hence, if all (or none) of the "T-events"
in the time series would coincide with "ET-events", then the average coincidence rates would be given by $r_{VAC_b} = 1$ (or $r_{VAC_b} = 0$). For independent time series, i.e. no coupling, $r_{VAC_b}$ would approximate the occurrence rate of "ET-events" in the time series (defined for $VAC_b$) that is governed by the chosen threshold, i.e. $r_{VAC_b} = \frac{1}{N} \sum_{i=1}^{N} 1_{[x_i^{ET} > th_{upper}^{ET}]}(x_i^{ET})$ (hence, $r_{VAC_b} \approx 0.3$ in our case). Coincidence rates $r_{VAC_c}$ follow equivalently by replacing $VAC_b$ with $VAC_c$ and in the definition of "ET-events" in the previous description. We compute $r_{VAC_b}$ and $r_{VAC_c}$ for all seasons but with an emphasis on
the warmest season of the year. In this study, significance of coincidence rates is established by randomly permuting one time series with respect to the other 100 times. Hence, $VAC$-rates from models or observations-based benchmarks that fall outside the 5th to 95th percentile range of the $VAC$-rates obtained from randomly permuted time series are significantly different from independent data at the 0.1 level.

In other words, $r_{VAC_b}$ gives the fraction of the highest 30% of temperatures that coincide with the highest 30% of ET (i.e.,
occurrence rate of 'energy-limited regimes'), while $r_{VAC_c}$ denotes the fraction of the highest 30% temperatures that correspond with the lowest 30% ET (i.e., occurrence rate of 'water-limited regimes'). Fig. 1c-d shows a simple example of monthly time series of T and ET simulated from two CMIP5 models and occurrences of $VAC_b$ and $VAC_c$ are highlighted, and Fig. 1e-f shows the correlation of T and ET. Note that for the same region (area-average over Central Europe, CEU) and time of the year (monthly data for June, July, and August), one model produces predominantly energy-limited regimes ($VAC_b$, Fig. 1c,e and
compare to conceptual illustration in Fig. 1a), whereas the other model produces predominantly water-limited regimes ($VAC_c$, Fig. 1d,f and concept in Fig. 1b).

We abbreviate the average occurrence rates $r_{VAC_b}$ and $r_{VAC_c}$ as $VAC_b$ and $VAC_c$ for convenience in the remainder of the paper. In comparison to more traditional coupling metrics, such as e.g. $\rho_{(T,ET)}$, $VAC$ might be expected to yield similar results

on very long time scales, whereas on shorter time scales the $VAC$ index picks up non-linearities in the tails (e.g. during warm temperature anomalies). At the monthly time scale (as used in this study), $VAC_b$ and $VAC_c$ detect distinct non-linearities in models and observations in summer T-ET coupling e.g. in CEU: Fig. S3 shows that, by correlating $VAC_b$ with $VAC_c$ derived from individual models, observations-based benchmarks, and from a two-dimensional Gaussian distribution, $VAC_b$ and $VAC_c$ rates in models and observations-based benchmarks exceed those that would be expected in random data. This deviation indicates that the warm tail is indeed different to the remainder of the distribution (we observe no such deviation for the cold tail, Fig. S3), and hence an evaluation metric that focuses on the tail such as the $VAC$ index is indeed useful for our present purpose. In addition to the main text, the model evaluation is presented for $\rho_{(T,ET)}$ to demonstrate robustness to the chosen methodological approach (Fig. S4), and for the $VAC$ index using a 90th percentile threshold (Fig. S4). Both alternatives show qualitatively similar results (see Results and Discussion section).

**A constraint on T-ET coupling in multi-model ensembles**

In general, a constraint links an observations-based diagnostic with a key model output variable across multiple models (Cox et al., 2013), and thus can be used to reduce model uncertainties and spread. Here, we derive a T-ET coupling constraint as the uncertainty range from the 54 combinations of T-ET benchmarking datasets. A Gaussian kernel with reliable data-based bandwidth selection (Sheather and Jones, 1991) is fitted over all 54 1989-2005 coincidence rates ($r_{VACc}$) for each meteorological season and pixel (and each SREX region average). Throughout this paper, the 5th to 95th percentile range of the fitted Gaussian kernels is taken as the plausible range of observations, and the reduced (constrained) ensemble of CMIP5 simulations is obtained by retaining only those CMIP5 models that simulate T-ET coincidences that fall within this range of observational uncertainty.

## 3   Results and Discussion

In this section, we first evaluate land-coupling in CMIP5 models explicitly against an observations-based ensemble of T-ET combinations and explore the link to temperature variability and extremes (Section 3.1). All model evaluation results are presented globally and exemplarily for Central Europe (CEU) as a region where global models and observations differ widely. Subsequently, we constrain the ensemble of CMIP5 models using each model's land-coupling as diagnosed through the $VAC_c$ index and discuss implications for biases in simulated present-day temperature extremes and warming projections (Section 3.2).

### 3.1   Evaluation of land-atmosphere coupling in CMIP5 models and the link to temperature variability and extremes

**Evaluation of T-ET coupling in CMIP5 models.**

Models and observations-based datasets show a relatively large spread in their representation of T-ET coupling, as expressed exemplarily in CEU through both $VAC_b$ and $VAC_c$ across various seasons (Fig. 2a,b) or diagnosed through more traditional coupling metrics such as $\rho_{(T,ET)}$ (Fig. S4). Individual models indicate pronounced qualitative differences in the warm season, where some models point to energy-limited, whereas others indicate predominantly water-limited conditions (Fig. 2a,b, and Fig. 1, for an illustrative example). Observations-based T-ET datasets agree qualitatively, i.e., indicating energy-limited to neutral conditions in the CEU example, thus implying an overestimation of water-limited regimes in CEU in roughly 50% of CMIP5 models (Fig. 2).

This pattern holds across most regions of the globe, as many CMIP5 models consistently overestimate occurrences of $VAC_c$ regimes (and correspondingly underestimate $VAC_b$ occurrences) in the warm season of the year (Fig. 2c,d, see Fig. S5 for a definition of the warm season in each pixel). In mid-latitude and several tropical regions (e.g. Central North America, Central Europe, the Amazon, India, parts of Africa), more than 25% and up to 50% of CMIP5 models lie outside the observational range (Fig. 2d). These discrepancies hold also if metrics that emphasize the whole distribution ($\rho_{(T,ET)}$) or more extreme parts of the tail (VAC based on a 90th percentile threshold) are used for model evaluation (Fig. S4, results for individual seasons are presented for $VAC_c$ and $VAC_b$ in Fig. S6 and Fig. S7, respectively). Moreover, the spread between the individual models' representation of land-atmosphere coupling strongly exceeds the spread in observational datasets, although different diagnostic, reanalyses and land surface model datasets are included in the observations-based ensemble (Fig. 2e for CMIP5 model spread and Fig. 2f for spread in observations-based benchmark datasets).

Furthermore, the models' land-atmosphere coupling, as diagnosed here through the VAC-index, is a highly model-inherent feature, as different model variants or ensemble members from the same model generally lie relatively close to each other (Figs. S1-S2). However, model-specific signatures of model output are not unusual, as diagnosed before for spatial patterns of temperature and precipitation (Knutti et al., 2013) or the statistical information content in carbon fluxes (Sippel et al., 2016a). Furthermore, present-day land-atmosphere coupling is strongly related to future land-atmosphere coupling in the individual models (Fig. S1). A detailed overview of $VAC_c$ coupling in individual models and ensemble members relative to the benchmark datasets for Central Europe and Central North America is presented in Fig. S1-S2. Despite regionally pronounced

qualitative discrepancies, it should be noted that on a global scale, the distribution of water-limited and energy-limited patterns in models and observations agrees qualitatively (Fig. S8). Likewise, the findings of climatologically too pronounced water-limited regimes in individual models w.r.t. observations does not exclude the possibility of future changes in the coupling strength in transitional regions (Seneviratne et al., 2006) or of strong water limitations during extreme events in the real world (Miralles et al., 2012; Whan et al., 2015). To this end, an evaluation of the year-to-year variability of the coupling behaviour in larger ensembles of individual models, including very rare events, could constitute a topic for further study, as this study was restricted to relatively moderate events in a 16 year period (70th percentile threshold for the computation of the VAC-index) and one ensemble member per model. Besides, we also note that observations-based benchmark datasets show systematic (albeit smaller) differences in the representation of land-atmosphere coupling: Diagnostic datasets indicate more frequent energy-limited regimes (see e.g. Fig. 2), and thus differ consistently to generally drier land surface models and reanalysis products, consistent with earlier findings (Santanello et al., 2015).

**T-ET coincidences and the link to temperature variability and extremes.**

The representation of T-ET coupling as diagnosed through the VAC index largely determines the variability of temperatures at monthly and inter-annual time scales across the CMIP5 multi-model ensemble in CEU (Fig. 3a) and in most regions of the globe except in some subarctic climates (Fig. 3b). Therefore, this relationship is indicative for the strong influence of land-atmosphere coupling on surface climate. This is consistent with previous findings in Europe in models with and without land-atmosphere interactions (Seneviratne et al., 2006; Fischer and Schär, 2009; Fischer et al., 2012). An important result is that models that produce $VAC_c$ indices within the range of benchmark datasets also produce a realistic near surface temperature variability, whereas models that fall too frequently in water-limited regimes also overestimate summer temperature variability (Fig. 3a). Moreover, in mid-latitude and tropical regions, the state of the land surface is strongly associated with the mean and variability of temperature extremes at the daily time scale in the warmest season (TXx, Fig. 3c,d). The link between the representation of land-atmosphere coupling and simulated temperature extremes and variability in global climate models is consistent with earlier studies, which has been demonstrated for Europe in individual models (Seneviratne et al., 2006; Lorenz et al., 2012; Davin et al., 2016) and in ensembles of regional models (Fischer et al., 2012; Bellprat et al., 2013). Therefore, the relationship between T-ET coincidence rates and temperature extremes might offer an avenue to derive an explicit land-atmosphere coupling constraint (the likely root cause for biases) to alleviate biases in temperature variability and extremes in the multi-model CMIP5 ensemble.

**3.2    Analysis of constrained multi-model ensemble and implications for future climate projections**

**A constraint on land-atmosphere coupling in the CMIP5 ensemble.**

The association between T and ET in the constrained ensemble resembles the observations-based benchmarking datasets in T-ET coupling very well (shown as a bivariate density estimate in Fig. 4a-b for CEU and CNA, respectively), whereas the unconstrained CMIP5 ensemble produces too many occurrences of $VAC_c$ conditions in both CEU and CNA. Due to the

intimate link between land-atmosphere coupling and temperature variability and extremes (see previous Section), we expect that the improvement in the representation of land-atmosphere coupling in the constrained ensembles yields a corresponding improvement in the representation of temperature extremes at the daily time scale in coupling-sensitive regions.

Coupling-sensitive regions are prone to warm season biases in climate models (Christensen and Boberg, 2012; Bellprat et al., 2013). In the present analysis, high biases in temperature extremes are indeed prevalent in the original (unconstrained) CMIP5 ensemble in these regions (Fig. 4c,e). For example, the ensemble mean warm season TXx is overestimated by up to 5°C, and higher biases are detected in the 90th percentile of TXx in CNA, CEU or the Amazon (all biases in daily variables relative to ERA-Interim, see Fig. 4c,e). In a CMIP5 ensemble constrained by the land-atmosphere coupling metric $VAC_c$, the representation of temperature extremes is improved in regions prone to coupling-induced biases (Fig. 4d,f), i.e. both mean TXx and the 90th percentile of TXx are significantly reduced. The ensemble mean of present-day temperature extremes in other regions remains unchanged. Moreover, projected future temperature extremes are reduced in the constrained ensemble (Fig. 5), similarly to present-day reductions in regions prone to present-day biases in land-atmosphere coupling. This is illustrated in Fig. 5a for TXx (monthly area-averages in summer) in CEU, where the hot end of the original model ensemble is in fact never realised in observed temperatures. The application of the constraint thus not only affects mean TXx, but also reduces the spread of the model ensemble (Fig.5a,b). The reduction in ensemble mean and ensemble spread is retained for the entire 21st century (Fig.5a,b). Hence, this result reinforces that coupling-related biases are model-inherent features, i.e. models that simulate too many $VAC_c$-occurrences today (and associated high biases in extreme temperatures) are very likely to do so in the future. However, one should keep in mind that the reduction in ensemble mean and spread is confined to coupling-sensitive regions in CEU, CNA, and to some degree in the Amazon region (Fig.5c,d).

Our results imply that an accurate representation of land surface processes is crucially relevant for a correct simulation of temperature extremes, and more generally for simulated near-surface climate variability. Land-atmosphere coupling is thus an important source of bias in state-of-the-art global climate model simulations. By using an observations-based land-atmosphere coupling diagnostic to constrain the multi-model CMIP5 ensemble, we have shown that biases in extremes in the large ensemble can be alleviated to a certain degree. As bias correction methodologies that take the physical causes for biases into account are still widely lacking (Ehret et al., 2012; Bellprat et al., 2013) and multivariate bias correction methods are currently in development (Cannon, 2016), the identification of models with a *physically plausible* representation of near-surface climate and land-atmosphere interactions at the regional scale might be crucial to extract accurate and relevant information about climate extremes in the context of climatic changes in the 21st century (Mitchell et al., 2016b; Schleussner et al., 2016; Seneviratne et al., 2016). For example, model selection for event attribution studies or a quantification of changes in univariate climate extremes is often based on a statistical performance criterion (Perkins et al., 2007; King et al., 2016; Otto et al., 2015). Our results indicate that these procedures could be further refined through incorporating observations-based diagnostics or constraints in order to analyse model simulations that are indeed "right for the right reasons" (at least given physics-guided and observations-based relationships). Moreover, the impacts of climate and its extremes e.g. on human health or ecosystems (Mitchell et al., 2016a; Frank et al., 2015) are often inherently related to multiple climate variables (Ehret et al., 2012; Leonard et al., 2014). Therefore, simple constraints as motivated for instance in the present study might complement more conventional

bias correction procedures (e.g. Hempel et al., 2013) to derive physically consistent estimates of climate impacts. This approach appears promising, because biases within climate models (i.e. in different variables) and across climate model ensembles are often correlated (e.g. Knutti, 2010; Mueller and Seneviratne, 2014; Sippel et al., 2016b). Hence, beyond soil moisture control on simulated temperature extremes as the present study's focus, related biases in other variables such as warm season precipitation

or ET might be similarly relevant in this context. For example, $VAC_c$ occurrences across the CMIP5 ensemble are negatively associated with precipitation and ET in the warm season in mid-latitude regions (Fig. S9) - both crucial variables in the water cycle that show pronounced summer low biases in CMIP5 models (Mueller and Seneviratne, 2014). Therefore, a constrained model ensemble with improved land-atmosphere coupling, a likely root cause of biases (Lorenz et al., 2012), might not only improve temperature extremes and variability, but additionally might reduce biases in associated variables such as ET or

precipitation.

**Is there a link between present-day land-atmosphere coupling and warming projections?**

We investigate whether the representation of land-atmosphere coupling in climate models affects the magnitude of 21st century warming (e.g. Fischer et al., 2012; Stegehuis et al., 2013). We first note that regions sensitive to land-atmosphere coupling in the CMIP5 model ensemble also show relatively strong warming in daily-scale temperature extremes (TXx), for example Central

America or South and Central Europe (Fig. 6a,b). More importantly, however, models that produce frequent $VAC_c$ occurrences (water-limited regimes) tend to be associated with larger rates of warming in TXx, although it should be emphasized that this relationship is not simple or linear (Fig. 6c,d, see also Fischer et al. (2012)). Conversely, this pattern reverses in boreal regions, where strongly energy-limited models (i.e. very few $VAC_c$ occurrences) tend to produce larger warming. However, in boreal regions this apparent relationship likely stems from a spurious correlation with the individual models' background warming

(i.e., warming in annual averages), as the correlation in fact disappears if the background warming is subtracted from summer warming (Fig. S10). In contrast, in mid-latitude regions warm season warming that exceeds annual average warming remains confined to the warm season. A multi-model projection constrained by a plausible representation of land-atmosphere coupling reduces differences in TXx estimates in a future climate relative to the present in coupling-sensitive regions such as Central Europe and Central North America by locally by around 0.5°C to 1°C - but this remains a regional effect (Fig. 6e,f). These

results are consistent with earlier studies that used an ensemble of regional models over Europe that used the standard deviation of temperatures as a constraint (Fischer et al., 2012).

## 4  Conclusions

In the present study, we have evaluated land-atmosphere coupling in state-of-the-art climate models with an ensemble of observations using a diagnostic based on coincidences of T and ET anomalies (the so called $VAC$ index). While observations and models broadly agree on spatial patterns of land-atmosphere coupling, our results reveal that models differ widely in coupling-sensitive regions in the mid-latitudes and the tropics. Several models exhibit systematically too frequent coincidences of high temperature anomalies with negative ET anomalies (water-limited regimes) in mid-latitude regions in the warm season, and in several tropical regions year-round. Across the multi-model ensemble, we found a strong association of land-atmosphere coupling with simulated temperature variability and extremes. The spread between models largely explains differences in simulated monthly temperature variability and daily extremes. We applied a land-atmosphere coupling constraint to the multi-model ensemble, which considerably improves the representation of land-atmosphere coupling in the ensemble, and reduces biases in temperature variability and extremes in present-day simulations in a physically consistent manner (Fig. 4). Furthermore, the constraint leads to reduced variability and lower extreme temperatures in future projections. However, the overall projected changes in temperature extremes are not so strongly affected (reduction around $0.5 - 1.0°$C locally in regions that are sensitive to land-atmosphere coupling), because the models with overestimated land-atmosphere coupling display similar anomalies from the multi-ensemble mean in present and future. In conclusion, we selected models with a *physically plausible* representation of land surface processes (and near-surface climate) using observations-based constraints that are guided by physical considerations. This approach complements more traditional bias correction approaches and offers new avenues to obtain improved estimates of future climate impacts.

*Acknowledgements.* We are grateful to the creators, maintainers, and providers of all data sets. CRU data were obtained from the University of East Anglia Climate Research Unit (CRU), British Atmospheric Data Centre, 2008, available from http://badc.nerc.ac.uk/data/cru. Furthermore, we acknowledge the Global Modeling and Assimilation Office and the Goddard Earth Sciences (GES) Data and Information Service Center (DISC) for the dissemination of MERRA, MERRA-LAND, and GLDAS-2 products (GL-NOAH-PF), the latter of which were acquired as part of the mission of NASA's Earth Science Division. The CFSR data are from the Research Data Archive which is maintained by the Computational and Information Systems Laboratory at the National Center for Atmospheric Research (NCAR). NCAR is sponsored by the National Science Foundation. The original data are available from the RDA (http://dss.ucar.edu) in data set number ds093.0. We would further like to acknowledge the Japanese 25yr ReAnalysis and JMA Climate Data Assimilation System (JCDAS) for the dissemination of JRA-25 data. The LandFlux-EVAL dataset used in this article was coordinated under the Global Energy and Water Exchanges (GEWEX) LandFlux initiative. We thank the World Climate Research Programme's Working Group on Coupled Modelling, which is responsible for CMIP, and we thank the climate modelling groups for producing and making available their model output. For CMIP the U.S. Department of Energy's Program for Climate Model Diagnosis and Intercomparison provides coordinating support and led development of software infrastructure in partnership with the Global Organization for Earth System Science Portals. We also thank Urs Beyerle and Jan Sedlacek for preparation and maintenance of CMIP5 data. This research has received funding from the European Commission Horizon 2020 BACI Project (Towards a Biosphere Atmosphere Change Index, grant No. 640176), and the European Research Council (ERC) under grant agreement No. 617518 (DROUGHT-HEAT). This work contributes to the World Climate Research Programme (WCRP) Grand Challenge on Extremes. S.S. is grateful to the German National Academic Foundation for PhD funding.

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

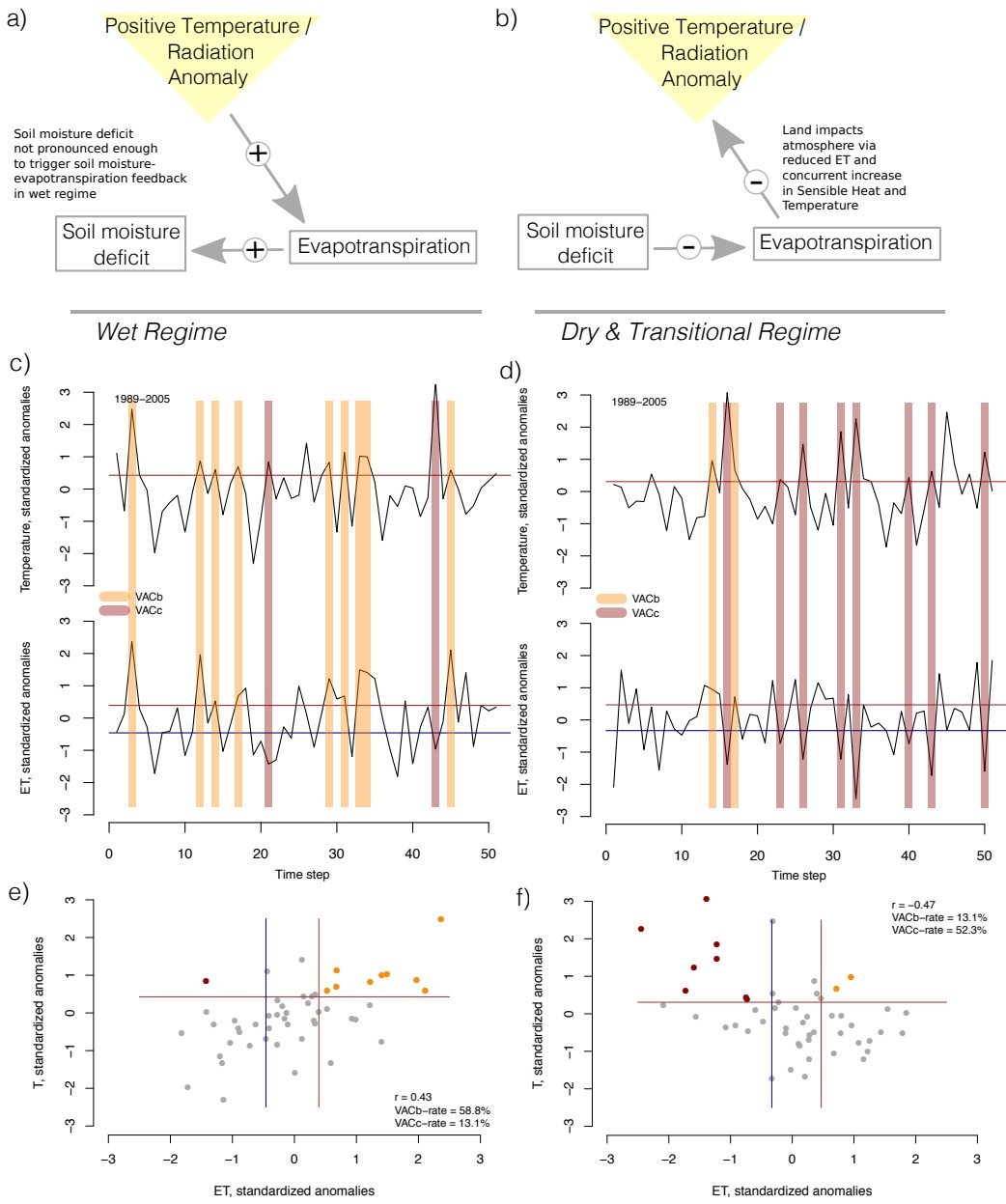

**Figure 1.** Illustration of qualitatively contrasting warm season temperature-evapotranspiration (T-ET) coupling in global climate models. (a, b) Conceptual illustration of T-ET coupling in (a) wet, and (b) dry & transitional regimes. In wet regimes T and ET are positively associated (atmosphere impacts land), while in dry & transitional regimes T and ET are negatively associated due to soil moisture feedbacks (i.e., land impacts atmosphere via reduced ET amd concurrent increases in sensible heat and T). (c-f) Different CMIP5 models show contrasting T-ET coupling behaviour in a mid-latitude region in summer (Central Europe, spatial average, JJA, 1989-2005): (c,e) NorESM1-M produces predominantly wet regimes, i.e. a positive T-ET coupling, while (d,f) ACCESS1-3 produces predominantly dry regimes (negative T-ET coupling), illustrated as time series (c-d) and in the T-ET plane (e-f). Red lines in (c-f) indicate $th_{upper}$ for $T$ and $ET$, blue lines indicate $th_{lower}^{T}$ (70th and 30th percentile in each individual time series, respectively).

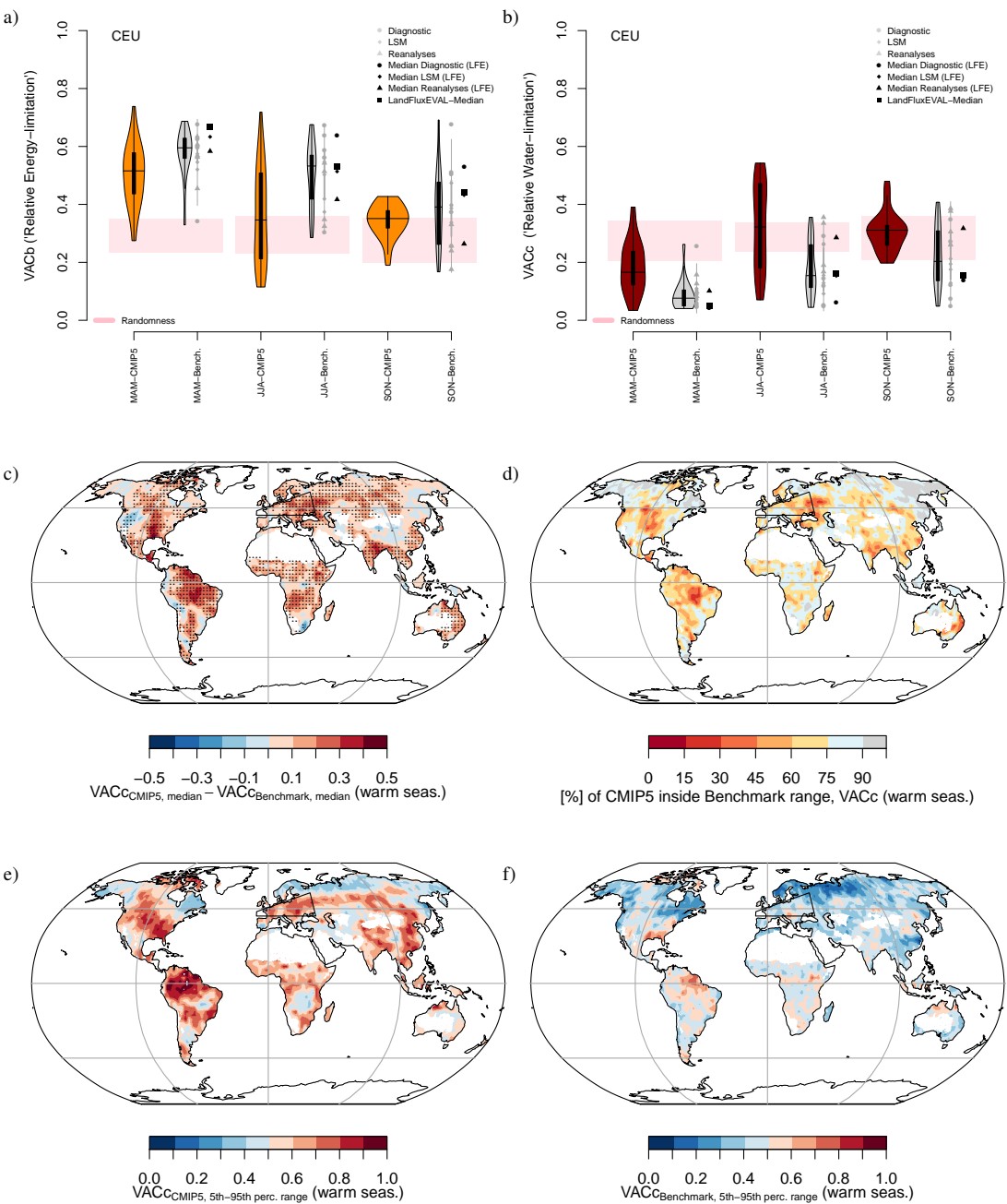

**Figure 2.** Evaluation of T-ET coupling in global climate models. (a, b) VACb and VACc coupling in the CMIP5 climate model ensemble and observations-based benchmarking datasets in Central Europe (CEU, 1989-2005, area-average) with systematic warm season differences (circles, diamonds, and triangles indicate diagnostic, land surface models, and reanalyses reference datasets, respectively). Randomness indicates the 5th to 95th percentile range obtained by randomly permutating both time series with respect to the other ($N = 100$ times) to obtain independent data. (c) Difference in the VACc median of the CMIP5 ensemble and benchmarking datasets. (d) Fraction of CMIP5 models that are inside the 5th-95th percentile spread of the benchmarking datasets. (e, f) Range of VACc-occurrences (5th to 95th percentile range) in CMIP5 models (e) and in the ensemble of observations (f). **21**

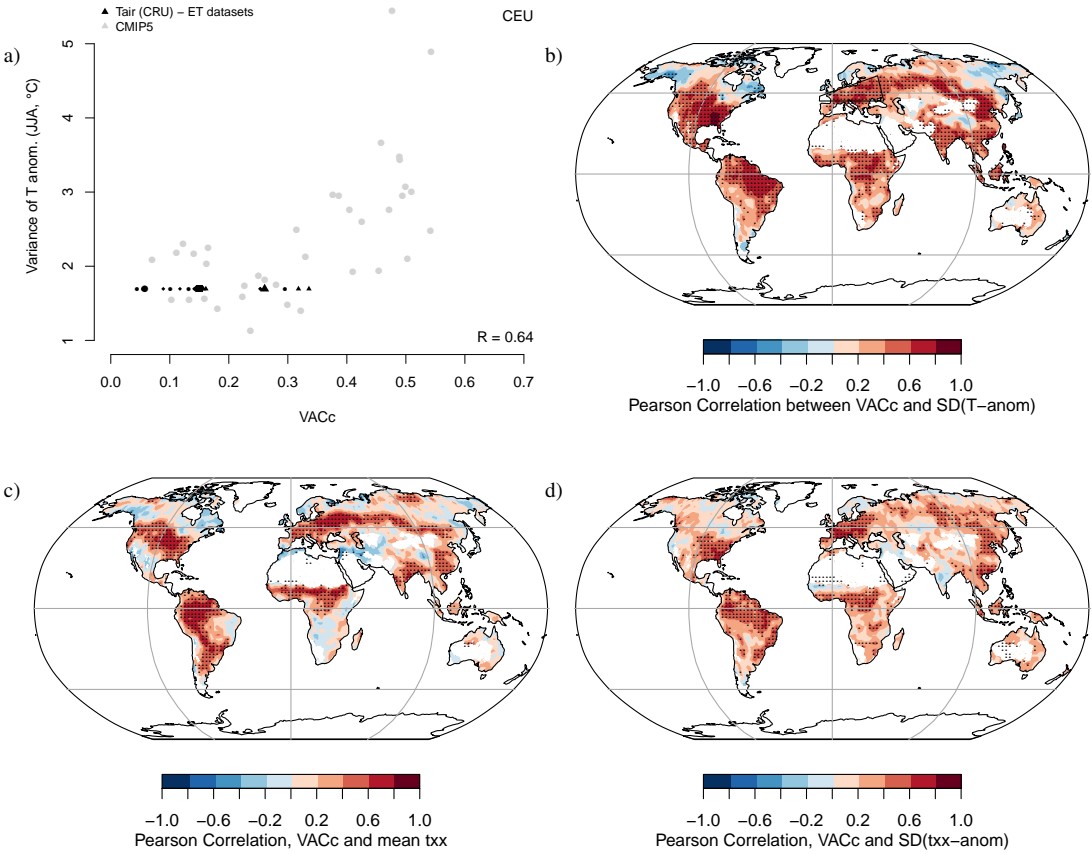

**Figure 3.** (a, b) Relationship between model-specific T-ET coupling (expressed through VACc) and model simulated variability of monthly temperature anomalies (JJA) in Central Europe (a), and globally (b). (c, d) Relationship betweeen VACc-coupling and mean (c) and standard deviation (d) of simulated monthly maximum value of daily maximum temperature (TXx) in summer (JJA).

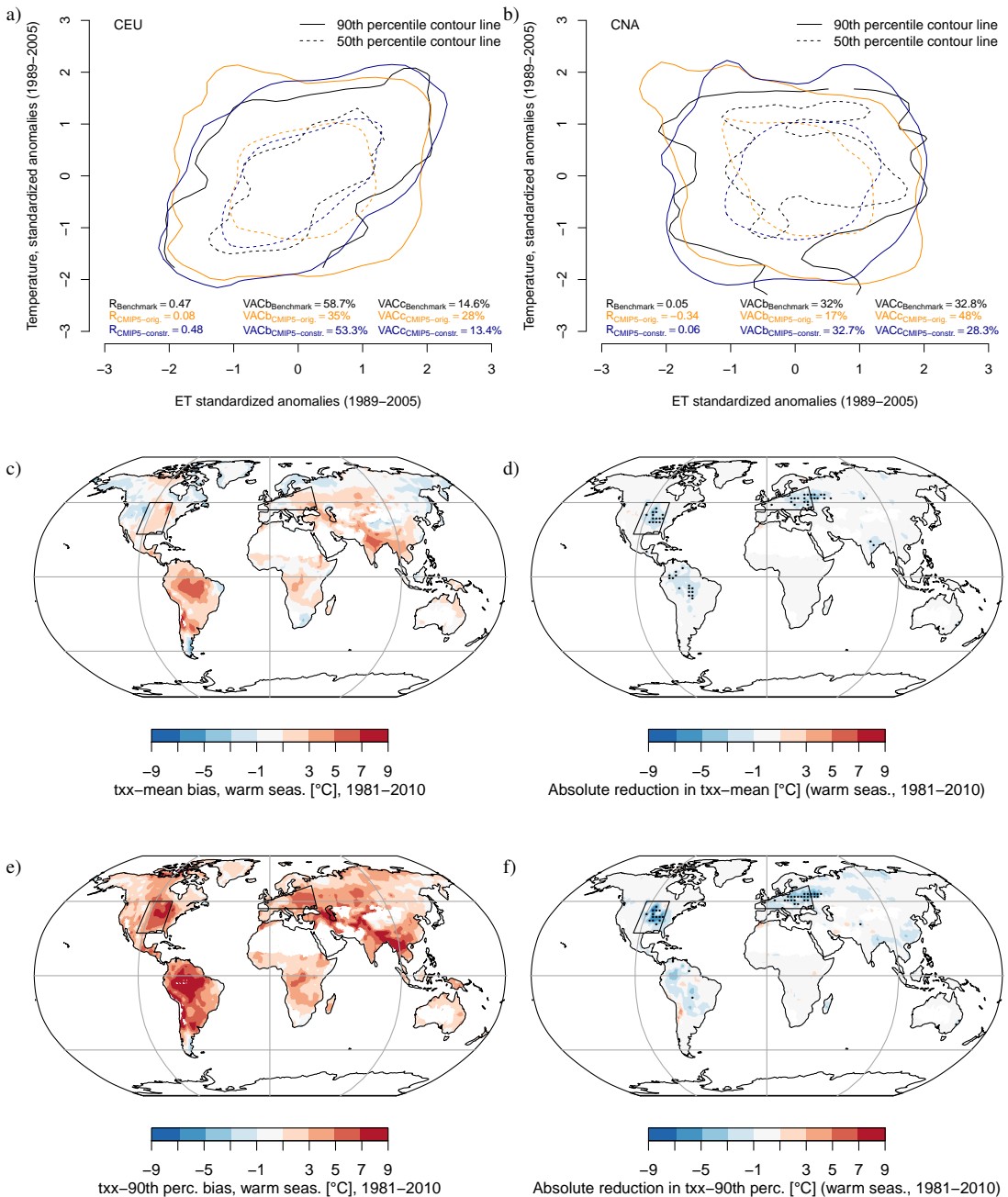

**Figure 4.** (a-b) Contour lines of bivariate kernel density estimates of T-ET relationship in the benchmarking datasets, the original and constraint CMIP5 ensemble for (a) Central Europe, and (b) Central North America (1989-2005, area-average). (c, e) Biases in warm season (c) TXx mean, and (e) 90th percentile of TXx in the original CMIP5 ensemble, and (d, f) reduction in (d) TXx mean, and (f) 90th percentile TXx through the application of the land-coupling constraint. Regions with a significant reduction in (d) TXx mean, and (f) the across-model average in the 90th percentile of TXx according to a permutation significance test are stippled.

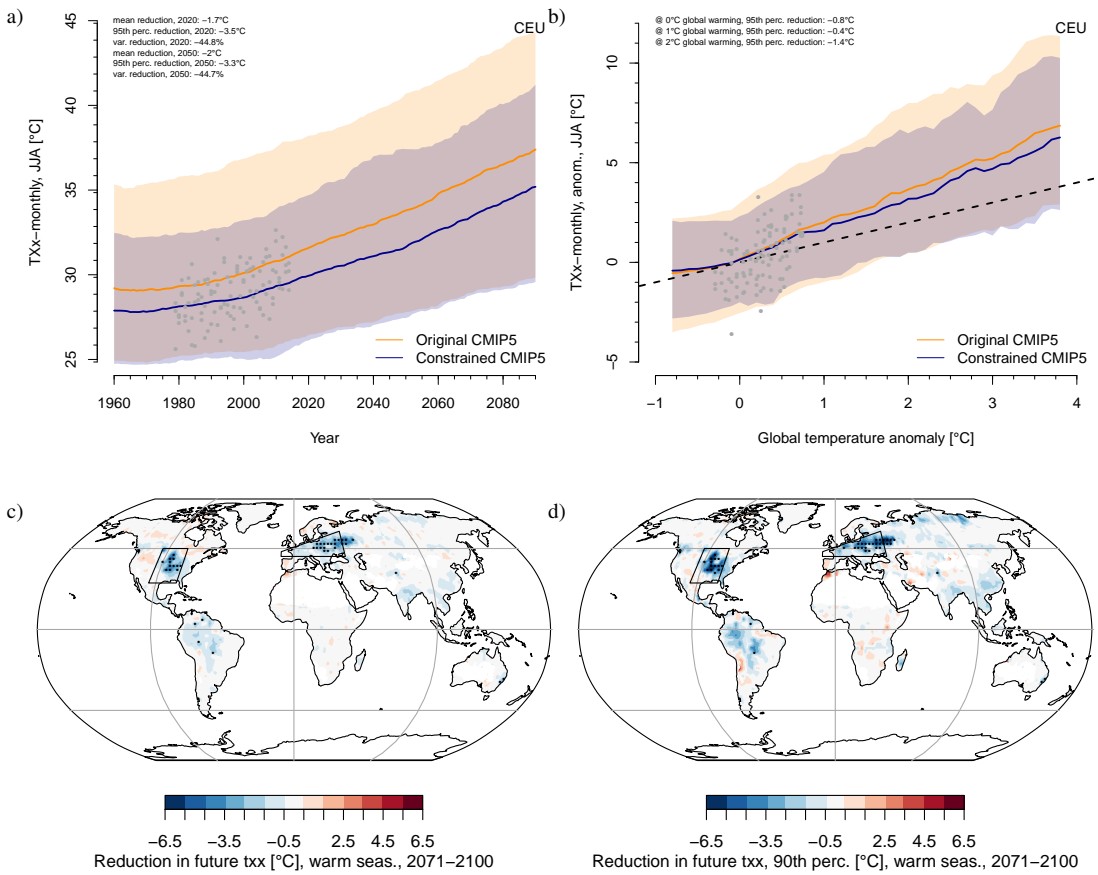

**Figure 5.** Application of land coupling constraint to CMIP5 ensemble. (a, b) Ensemble prediction of original and constrained multi-model ensemble for (a) future absolute TXx and (b) range of TXx anomalies relative to global mean temperature anomalies in each model, following Seneviratne et al (2016). Envelopes indicate 5th to 95th percentile. (c, d) Global maps of projected changes in simulated (c) mean TXx, and (d) 90th percentile of TXx in the VACc-constrained CMIP5 ensemble.

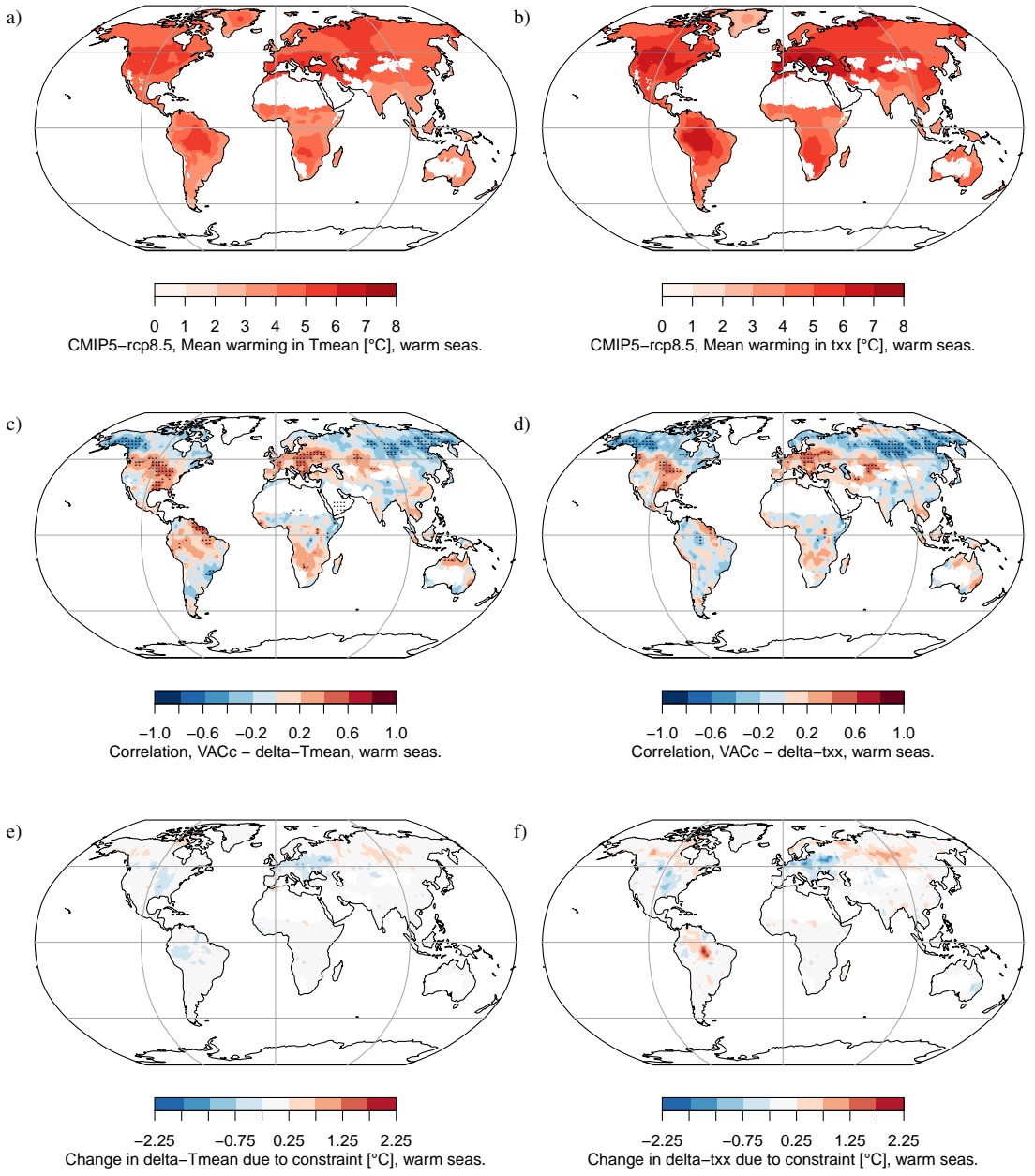

**Figure 6.** (a, b) Projected warming in warm season (a) mean temperature, and (b) TXx across the CMIP5 ensemble (RCP8.5 scenario, 2071-2100 relative to 1981-2010). (c, d) Correlation between VACc in the warm season and the projected warming in (c) mean temperature, and (d) TXx. Stippling indicates significant correlations. (e, f) Relative change in (e) mean warming and (f) TXx warming due to the application of the land-atmosphere coupling constraint, warming defined as 2071-2100 relative to 1981-2100.

**Table 1.** Datasets used for model evaluation

| Name of dataset | Variable | Type / Group | Provider & Reference |
|---|---|---|---|
| LandFlux-EVAL[a] | ET | Ensemble Median | Mueller et al. (2013) |
| LandFlux-EVAL[a] | ET | Median of Reanalyses | Mueller et al. (2013) |
| LandFlux-EVAL[a] | ET | Median of LSMs | Mueller et al. (2013) |
| LandFlux-EVAL[a] | ET | Median of Diagnostic datasets | Mueller et al. (2013) |
| PRUNI[a,b] | ET | Diagnostic | Sheffield et al. (2006, 2010) |
| MPIBGC[a,b] | ET | Diagnostic | Jung et al. (2011) |
| CSIRO[a,b] | ET | Diagnostic | Zhang et al. (2010) |
| GLEAM[a,b], V. 1A | ET | Diagnostic | Miralles et al. (2011a, b) |
| AWB[a,b] | ET | Diagnostic | Mueller et al. (2011a) |
| EI-ORCHIDEE[a,b] | ET | LSM | Krinner et al. (2005) |
| CRU-ORCHIDEE[a,b] | ET | LSM | Krinner et al. (2005) |
| VIC[a,b] | ET | LSM | Sheffield et al. (2006); Sheffield and Wood (2007) |
| GL-NOAH-PF[a,b] | ET | LSM | Rodell et al. (2004); Rui and Beaudoing (2016) |
| MERRA-LAND[a,b] | ET | LSM | Reichle et al. (2011) |
| ERA-Interim[a,b] | ET | Reanalysis | Dee et al. (2011) |
| CFSR[a,b] | ET | Reanalysis | Saha et al. (2010) |
| JRA-25[a,b] | ET | Reanalysis | Onogi et al. (2007) |
| MERRA[a,b] | ET | Reanalysis | Bosilovich (2008) |
| CRU-TS3.2[a] | T | Observations | Harris et al. (2014) |
| ERA-Interim reanalysis[a] | T | Reanalysis | Dee et al. (2011) |
| CFSR reanalysis[a] | T | Reanalysis | Saha et al. (2010) |

[a] All T-ET combinations of marked datasets have been used to derive the ET-T constraint.

[b] Original individual datasets that contributed to the LandFlux-EVAL synthesis project (Mueller et al., 2013).