# Peer review of "Refining multi-model projections of temperature extremes by evaluation against land-atmosphere coupling diagnostics"

_Earth System Dynamics, 2016_

## Referee Comment (RC1) · Anonymous Referee #1 · 20 Dec 2016

The paper "Refining multi-model projections of temperature extremes by evaluation against land-atmosphere coupling diagnostics" uses temperature and ET to explore the land-atmosphere interactions during heat waves in models. It then uses the derived coupling metric to constrain the CMIP-5 models and show that the constrained ensemble provides better representation of heat waves. Overall the paper is interesting and should be published, however it is poorly written in that there is insufficient justification of the methodology, overuse of supplementary figures and a lack of discussion and analysis. This makes the paper difficult to follow and understand. The authors present too much material and they need to trim it down to provide a strong and concise message. Therefore, I am recommending that the paper needs major revisions

before it is published. The specific comments that motivated this decision are given below.

Specific Comments:

Pg. 2, lines 3-7: This description makes sense but I cannot see the connection between this description and the and the "conceptual" Fig. 1a,b. It may be conceptual for the authors, but there needs to be more description about what each of the symbols in the figures means and how it connects to the description in the text. Specifically, what is the difference between a thick and thin arrow, curved versus straight, positive-negative sign, square, circle, rectangle and different colors. There is a lot going on in the figure and it is hard to know what is important and what is just there for aesthetic purposes.

Pg 4, line 12: Why use the old version of the NCEP reanalysis and not the CFSR? Why not use datasets with consistent temperature, evaporation datasets like MERRA and CFSR. These datasets have both temperature and evaporation. Why would you expect ET from one dataset to be correlated with T from another? There needs to be more discussion on this.

Pg 4, lines 19-20: Make sure to reemphasize this when discussing the results.

Pg 4, line 28: This is a good use of supplementary material as there is sufficient description to know that it contains a list of the 37 models used, but it is not necessary for understanding and interpreting the results of this paper.

Pg 4, lines 28-30: This is an important assumption for this study and the description is a bit vague. What does "tend" mean? It would be better to provide some sort of quantitative measure of the variability across ensembles. Is this true for all locations? Given the importance of this assumption there needs to be further analysis and discussion as to why you think this is a reasonable assumption.

Pg 6, line 2: I find this equation and description extremely confusing. It took me several times of reading the text to understand the metric. I am still have no idea of what is

represented in the equation particularly in the summation. Where does the 1 come from and why does it have a subscript? The equation is more confusing than the text and does not help at all with understanding the metric. From what I understand from the metric, the VACb gives the percent of the highest 30% of temperatures that correspond with highest 30% of ET, while VACc gives the percent of the highest 30% temperatures that correspond with the lowest 30% ET. It is difficult to remember which was VACb and VACc. It would be helpful if there was naming convention that is more descriptive instead of b and c.

Pg 6, line 10: I thought Fig 1a,b was just a conceptual example, how does this connect with the simple example of monthly time series referenced here? It seems to me that Fig 1a,b are completely unrelated with the rest of the figure, so why put them together? If they are related, then there needs to be more description as to how they are connected.

Pg 6, line 14-16: "Might" is not very reassuring and is an inherently weak argument. Correlation is also universally known and if you use a rank correlation it can also pick-up on the non-linearities.

Pg 6, lines 16-19: Not sure what this means. This needs more discussion. Also, this is an inappropriate use of supplementary information. There is no information about what is plotted in S1. Furthermore, this seems like an important justification of the methodology and should be included in the text.

Pg 6, lines 18-19: If it yields quantitatively similar results then why bother with the VAC? It is much easier to understand correlation. What about the significance level for the VAC? Is there any way to statistically quantify the significance of the VAC as to provide some level of confidence? If not, that is a major disadvantage over a traditional correlation metric and needs to be discussed. I am not against using the VAC, but as presented here the justification for using it is extremely weak. You need to convince the reader that this obscure metric is worth using.

Pg 6, line 20: This is a reasonable use of a supplementary figure because it provides more depth to the analysis, but is not directly necessary for understanding the paper. It could be improved by providing a better description. For example, "the model evaluation as shown in Fig X is presented for a 90th percentile threshold in Fig S2 and shows …" If this figure is completely different from any other figure presented in the text, then you need to include a more descriptive discussion about it in the text.

Pg 8, lines 4-5: A figure is not a reference, don't use it like one. There needs to be discussion about the figure and what it shows. Parts a and b need to be explained more. I am unsure as to what the different shapes represent and the colors.

Pg 8, line 10: Fig. 3b does not say Pearson correlation, make sure it is consistent with the other plots.

Pg 8, line 25: It is ok to reference a figure like this if you have already discussed it but since this is the first time that fig 4 has been mentioned you need to describe what is being plotted. Also, this is an inappropriate use of supplementary material. There needs to be explanation in the text. How does it differ from what is being plotted in fig 4a-b. Seems like the authors are using supplementary figures instead of actually discussing the important aspects of the analysis.

Pg 8, line 31: I think Fig. 4 is the best and most impactful figure in this paper. Make sure that you emphasize its importance.

Pg 9, line 2: What does "substantially" mean? Is it significant statistically speaking?

Pg 9, line 25: Again there is no information about what is plotted in the figure and the text makes it sound relevant for understanding and interpreting the results for this paper and therefore it should be included in the actual paper and not as supplementary material.

Pg 10, Line 3: Again, supplementary plots are not a reference. There needs to be an explanation of the figure.

Pg 10, Line 8: If you reference the same supplementary figure more than once, then that is a good indication that it should be included it in the text. There is more discussion in the text on S13 then on Fig. 5.

———————————————————

---

## Referee Comment (RC2) · Anonymous Referee #2 · 3 Feb 2017

The manuscript by Sippel et al. addresses the reduction of ensemble temperature projections by using best estimates of soil moisture-temperature coupling diagnostics under current climate conditions. Although the technique itself has been applied in several other studies, the current application is novel and the results are highly relevant for our understanding of projections of temperature extremes. The manuscript is generally well-written and results are presented in a concise manner. The work seems technically sound, and I could not detect any major flaws in the reasoning and/or analysis, although some minor points were identified that will need to be addressed. Therefore, I believe the manuscript can be accepted for publication after minor revisions.

[Figure]

I have the following remarks/observations:

- VAC is based on the 30/70th percentile, whereas the authors consider the 90th percentile of TXx. Please motivate if and why this is justified and consistent (coupling might be different for highest percentiles).

- While the manuscript has a balanced number of display items, I found the link between the information displayed and that discussed in the text weak. Many sub-panels are never mentioned or discussed, and too much is left for the reader to interpret. Please make sure all relevant information in the figures is referred to, as well as all figures and sub-panels themselves. In particular a more in depth-discussion of the results in Figures 5 and 6 is needed.

- The selection of references doesn't always to justice to work that other groups have been doing in this area or on this specific topic. In the introduction on weighing models in large ensembles (Page 3, lines 8–12), some examples are provide but interestingly the ones most relevant to the current work are not cited (i.e. Fischer et al., 2012 and Stegehuis et al., 2013). In this way, the suggestion is made that this study is the first to apply model selection on temperature extremes. Please include references to these works. Also, model selection/weighing has been applied to other aspects/fields such as snow albedo feedback (Hall and Qu, 2006) and hydrological drought projection (Van Huijgevoort el al., 2014). When discussing the vegetation-atmosphere coupling index (VAC), the authors refer to previous work from the group (e.g. Seneviratne et al., 2006; Lorenz et al., 2012) from which VAC was developed, but not to other alternative indices that are based on a similar philosophy (for instance the metric developed by Miralles et al., 2012, although this paper is cited in a different context).

**References**

Hall, A. & X. Qu (2006) Using the current seasonal cycle to constrain snow albedo feedback in future climate change. *Geophys. Res. Lett.*, 33, L030502, doi:10.1029/2005GL025127.

Van Huijgevoort, M.H.G., et al. (2014) Identification of changes in hydrological drought characteristics from a multi-GCM driven ensemble constrained by observed discharge. *J. Hydrol.*, 512, 421–434.

---

## Author Comment (AC1) · 12 Feb 2017

Thanks for constructive comments. Please find a detailed reply to all comments attached. Best Wishes, Sebastian Sippel on behalf of all authors

Please also note the supplement to this comment:
http://www.earth-syst-dynam-discuss.net/esd-2016-48/esd-2016-48-AC1-supplement.pdf

---

## Author Response (AR1)

| | |
|---|---|
| 1 | Jena, 07 March 2016 |
| 2 | |
| 3 | |
| 4 | Dear Editors and Reviewers, |
| 5 | |
| 6 | we thank both reviewers for their positive evaluation of our manuscript and their |
| 7 | constructive comments. We also appreciate the more critical comments that were |
| 8 | mainly related to structure and clarity of the manuscript. |
| 9 | We respond to all comments in a point-by-point manner in individual responses, which |
| 10 | are reproduced below and in few cases slightly expanded. Changes made to the |
| 11 | manuscript are highlighted in an additional file. |
| 12 | We believe that the manuscript has improved significantly on the basis of the comments, |
| 13 | in particular with respect to clarity, structure, balance of cited references, and in |
| 14 | providing a concise message. |
| 15 | Please do not hesitate to contact us in case any further questions arise. We would highly |
| 16 | appreciate if our manuscript could be considered suitable for publication in *Earth* |
| 17 | *System Dynamics*. |
| 18 | |
| 19 | Sincerely, |
| 20 | Sebastian Sippel on behalf of all authors |
| 21 | |

**Reply to Anonymous Referee #1 comment on Sippel et al., 2016, Earth System Dynamics Discussions, doi:10.5194/esd-2016-48**

The paper "Refining multi-model projections of temperature extremes by evaluation against land-atmosphere coupling diagnostics" uses temperature and ET to explore the land-atmosphere interactions during heat waves in models. It then uses the derived coupling metric to constrain the CMIP-5 models and show that the constrained ensemble provides better representation of heat waves. Overall the paper is interesting and should be published, however it is poorly written in that there is insufficient justification of the methodology, overuse of supplementary figures and a lack of discussion and analysis. This makes the paper difficult to follow and understand. The authors present too much material and they need to trim it down to provide a strong and concise message. Therefore, I am recommending that the paper needs major revisions before it is published. The specific comments that motivated this decision are given below.

**We thank the reviewer for the overall positive comments on our manuscript, and also acknowledge the more critical comments. We understand that the critical comments are mainly related to an indeed densely written manuscript, partly unsatisfactory description of figures/methods, and that the manuscript would benefit from a more concise message. In a revised version of the manuscript, we focus on the two main findings of this paper to provide a strong and concise message: Namely, we show that**

1) **a relatively large fraction of CMIP5 models (up to ~50% in some regions) misrepresent warm season land-atmosphere coupling in mid-latitude regions, i.e. the occurrence of water-limited regimes (coincidences of high T anomalies with low ET anomalies) are systematically overestimated relative to observations-based datasets;**
2) **the representation of land-atmosphere coupling (i.e. T-ET coincidences) in the models is closely related to biases in the simulation of temperature extremes; therefore the application of a "land-atmosphere coupling constraint" derived from our ensemble of observations-based datasets reduces biases in temperature extremes and variability in a physically consistent way.**

**Please note also that we have restructured the Abstract in that it now specifies the presentation of these two main findings in logical order. Also, we have reduced the number of Supplementary Figures to avoid unnecessary detail.**

**Furthermore, we improve the description of figures/methods, elaborate on explanations**
**and discussions as outlined below in response to the reviewer's specific comments.**

Specific Comments:

Pg. 2, lines 3-7: This description makes sense but I cannot see the connection between this
description and the "conceptual" Fig. 1a,b. It may be conceptual for the authors, but there needs
to be more description about what each of the symbols in the figures means and how it connects
to the description in the text. Specifically, what is the difference between a thick and thin arrow,
curved versus straight, positive-negative sign, square, circle, rectangle and different colors.
There is a lot going on in the figure and it is hard to know what is important and what is just
there for aesthetic purposes.

**We agree with the reviewer's critique, and provide a detailed description of the purpose**
**of all graphical elements in the revised manuscript. Specifically, we have removed curved**
**arrows, misleading colors and shapes in the revised figure (except that the yellow**
**triangle indicates an external trigger of feedbacks). Plus and minus signs indicate positive**
**and negative impacts, respectively. We hope all this becomes clear in the revised Figure**
**and explanation.**

Pg 4, line 12: Why use the old version of the NCEP reanalysis and not the CFSR? Why not use
datasets with consistent temperature, evaporation datasets like MERRA and CFSR. These
datasets have both temperature and evaporation. Why would you expect ET from one dataset to
be correlated with T from another? There needs to be more discussion on this.

**One of the main goals of our study was to explicitly use as many ET datasets as possible**
**(including diagnostic/empirical datasets derived from e.g. remote sensing or flux**
**measurements), because there remains a very large spread between individual ET**
**products (see e.g. Mueller et al., 2013). Therefore, using only "consistent" T-ET**
**combinations from a specific reanalyses would exclude diagnostic datasets and would not**
**sample the large spread between the different ET products. However, despite this all T-ET**
**dataset combinations are originally derived from real-world observations (or land**
**models driven by observations), and therefore can be expected to represent relevant**
**features in the observations, especially on large spatio-temporal scales (here monthly**
**data, mostly on SREX-region level). Please note also that the coincidence methodology is**
**robust to noise in that it is not sensitive to any numerical values or outliers in the data**
**(but e.g. covariance or correlation-based metrics would be sensitive to numerical values),**

but only considers threshold exceedances (see e.g. Donges et al. (2016) for an in-depth description of the method and Schleussner et al. (2016) for an application).

Most importantly, however, we have investigated the T-ET consistency issue in detail, and find that the spread in T-datasets is basically negligible in comparison to the large uncertainties and discrepancies in ET-datasets (please see figures 1 - 3 below):

Figure 1 below is reproduced from Fig. 2 in the main manuscript and shows the distribution of (a) VACb ("energy-limitation") and (b) VACc ("water-limitation") in CMIP5

models and various combinations of T-ET datasets. The ET datasets are derived through diagnostic means (circles), land surface model simulations (diamonds), and reanalyses (triangles). If we assume that an "inconsistent" combination of T and ET datasets (i.e.

from different sources) would not be feasible for the present analysis (as suggested by the

Reviewer), then we would expect these "inconsistent" T-ET combinations to be shifted towards independent noise without correlations (i.e. symmetrically around the pink range). However, to the contrary, we find that the observations-based dataset combinations are indeed not strongly affected by noise, as they lie systematically above (Fig. 1a) or below (Fig. 1b) the noise level. More importantly, however, we have tested explicitly for the effect of choosing different temperature datasets (see Figure 2 below).

Here, we find that changing the temperature observations that are to be combined with

ET datasets has only a minor effect on the location of the respective grey or black dot in the VACb-VACc diagram (e.g. three black triangles are the Median Reanalyses ET dataset combined with CRU-temperature, ERAI-temperature, and NCEP-temperature, respectively). This indicates that by far the largest source of uncertainty stems from the choice of ET dataset, and the uncertainty that stems from the choice of temperature dataset is (almost) negligible. Also, in Fig. 2 we also find that the observations-based T-ET

combinations are systematically shifted from the "random range" (around 0.3 in both

VACc/VACb) to favour the more water-limited (high VACb) regimes (except one diagnostic ET dataset that in many cases tends to lie around random and uncorrelated data). Finally, a direct comparison between T datasets reveals that on a monthly time scale these datasets are almost perfectly correlated (Figure 3, top). In contrast, correlations between different ET datasets are low (Figure 3, bottom), which indeed confirms that the uncertainty induced by the choice of T-dataset is almost negligible given the large uncertainties induced by the spread between different ET datasets.

We include more discussion on this issue in the revised manuscript (Section 2.1).

**Regarding temperature from NCEP/DOE reanalysis vs. CFSR reanalysis: Thanks for**

**pointing out that NCEP/DOE is an old version. We have replaced the NCEP/DOE reanalysis**

**by the newer CFSR reanalysis and redone all calculations with the three temperature**

**datasets CRU, ERAI, and CFSR (instead of CRU, ERAI, and NCEP/DOE previously). Results**

**are almost exactly the same (as to be expected due to well-correlated temperature**

**datasets, see above).**

[Figure]

Figure 1: Evaluation of T-ET coupling in global climate models. (a, b) VACb (a) and VACc
coupling in the CMIP5 climate model ensemble and observations-based benchmarking
datasets in Central Europe (CEU, 1989-2005, area-average) with systematic warm season
differences. Randomness indicates the 5th to 95th percentile range obtained by randomly
permutating both time series with respect to the other (N = 100 times) to obtain independent
data (reproduced from Fig. 2a,b in the main manuscript).

[Figure]

Figure 2: VACb plotted against VACc (Central Europe, JJA) for CMIP5 models and various T-ET combinations. All ET datasets have been combined with each CRU-temperature, NCEP-temperature, and ERAI-temperature, and therefore each dot is plotted three times.

[Figure]

Figure 3: Correlation between different Tair datasets (top), and different ET datasets (bottom).

Pg 4, lines 19-20: Make sure to reemphasize this when discussing the results.

**We reemphasize this point in the Results and Discussion section.**

Pg 4, line 28: This is a good use of supplementary material as there is sufficient description to
know that it contains a list of the 37 models used, but it is not necessary for understanding and
interpreting the results of this paper.

Pg 4, lines 28-30: This is an important assumption for this study and the description is a bit
vague. What does "tend" mean? It would be better to provide some sort of quantitative measure
of the variability across ensembles. Is this true for all locations? Given the importance of this
assumption there needs to be further analysis and discussion as to why you think this is a
reasonable assumption.

**We agree this is an important point that needs more clarification: VACc at present is a**
**very good predictor for VACc in the future (significant correlation globally across almost**
**all locations, see Figure 4 below). This indicates that variability in land-atmosphere**
**coupling across models as diagnosed by the VACc index is largely a model-inherent**
**feature that is determined by model structure, and model-internal variability on 30-year**
**time scales plays only a comparatively minor role. More importantly, Figures S6 and S7 in**
**the Supplementary Material of the Discussion version confirm this interpretation: While**
**there is some variation in VACc within individual models in the 1989-2005 period (mostly**
**in models with many ensemble members including perturbed physics), the large model**
**spread overall is clearly dominated by across-model variability (and therefore we have**
**chosen only one ensemble member per model -the often-chosen standard run r1i1p1- to**
**investigate variability *across* the CMIP5 ensemble without being confounded by inequal**
**ensemble sizes for individual models).**

**However, it is important to note that there might be substantial variability in land-**
**atmosphere coupling within models or ensemble runs of the same model for individual**
**(e.g. extreme) years (see e.g. Sippel et al., 2016 for a regional model over Europe) or on**
**multi-year but not climatological time scales, or for model ensembles with perturbed**
**physical parametrisations (which is beyond the scope of this manuscript but could be**
**promising future research). We clarify all these points in the revised manuscript.**

[Figure]

Correlation between VACc_present and VACc_future (CMIP5)

Figure 4: Correlation between present-day and future land-atmosphere coupling reveal that occurrences of VAC$_c$ are largely model-inherent features.

Pg 6, line 2: I find this equation and description extremely confusing. It took me several times of reading the text to understand the metric. I am still have no idea of what is represented in the equation particularly in the summation. Where does the 1 come from and why does it have a subscript? The equation is more confusing than the text and does not help at all with understanding the metric. From what I understand from the metric, the VACb gives the percent of the highest 30% of temperatures that correspond with highest 30% of ET, while VACc gives the percent of the highest 30% temperatures that correspond with the lowest 30% ET. It is difficult to remember which was VACb and VACc. It would be helpful if there was naming convention that is more descriptive instead of b and c.

**We agree that the equation is confusing and aim to explain it better in the revised manuscript. Your interpretation of the VACb and VACc metric is correct - and we will clarify this interpretation in the text (i.e. include a verbal description in the text after the equations).**

**The equation is simply there to state that we count coincidences of T and ET in a given category (e.g. positive T *and* negative ET) to get the average coincidence rate r_VACb. The "1" in the equation just means we count 1 for each occurrence of VACb - and 0 otherwise (the indicator function, https://en.wikipedia.org/wiki/Indicator_function).**

**Lastly, we also put more emphasis on the meaning of VACb ("energy-limitation") and**
**VACc ("water-limitation"). However, we would not rename VACb and VACc, just to be**
**consistent with earlier studies that introduced the VAC-metric (Zscheischler et al., 2015).**

Pg 6, line 10: I thought Fig 1a,b was just a conceptual example, how does this connect with the
simple example of monthly time series referenced here? It seems to me that Fig 1a,b are
completely unrelated with the rest of the figure, so why put them together? If they are related,
then there needs to be more description as to how they are connected.

**The subfigures are indirectly related: In Fig. 1a and Fig. 1b we conceptualize "wet" and**
**"dry" regimes, respectively. Fig. 1c/e and Fig. 1d/f then illustrate how time series and the**
**T-ET correlations look like in models that predominantly produce "wet" (NorESM1-M)**
**and "dry" (ACCESS1-3) regimes, respectively. In the revised version, we explain this point**
**better in the caption and in the text. The new text reads (i.e. Section 2.2):**

"Fig. 1c-f shows a simple example of monthly time series of T and ET simulated from two CMIP5
models and occurrences of VACb and VACc are highlighted. Note that for the same region (area-
average over Central Europe, CEU) and time of the year (monthly data for June, July, and
August), one model produces predominantly energy-limited regimes (VACb, Fig. 1c,e and
compare to conceptual illustration in Fig. 1a), whereas the other model produces predominantly
water-limited regimes (VACc, Fig. 1d,f and concept in Fig. 1b)."

Pg 6, line 14-16: "Might" is not very reassuring and is an inherently weak argument. Correlation
is also universally known and if you use a rank correlation it can also pick-up on the non-
linearities.

**"Might" can be safely removed because in the next sentence as we say (and show) that it**
**does pick up non-linearities (see next comment below).**

Pg 6, lines 16-19: Not sure what this means. This needs more discussion. Also, this is an
inappropriate use of supplementary information. There is no information about what is plotted
in S1. Furthermore, this seems like an important justification of the methodology and should be
included in the text.

**In the revised manuscript we provide an expanded explanation what we mean:**
**Essentially, correlations (Pearson, rank, etc.) emphasize the whole distribution, while our**
**coincidence analysis using VACb and VACconly looks at the warm tail. Figure S1 shows**

**that, by correlating VACb and VACc in both models (red), observations (black), and**
**artificial data sampled from a two-dimensional Gaussian distribution with different**
**covariances, both models and observations (red and black dots in bottom left subfigure in**
**S1) the VACb and VACc rates exceed those that would be expected in artificial data (grey**
**dots). This deviation indicates that the warm tail is indeed different to the remainder of**
**the distribution (e.g. no such deviation is detected for the cold tail, VACa and VACd in the**
**bottom right subfigure); and hence an evaluation metric that focuses on the (warm) tail is**
**indeed useful.**

Pg 6, lines 18-19: If it yields quantitatively similar results then why bother with the VAC? It is
much easier to understand correlation. What about the significance level for the VAC? Is there
any way to statistically quantify the significance of the VAC as to provide some level of
confidence? If not, that is a major disadvantage over a traditional correlation metric and needs
to be discussed. I am not against using the VAC, but as presented here the justification for using
it is extremely weak. You need to convince the reader that this obscure metric is worth using.

**Please consider a few things regarding the method:**

**First, as noted above, we believe that it is an advantage to use "non-traditional"**
**evaluation metrics in this case because these focus on the tails rather than the whole**
**distribution (and this would become even more important for sub-monthly or daily time**
**series).**

**Second, the method is robust and non-parametric - i.e. it is not sensitive to numerical**
**values or outliers in the data. For this reason it has been used in a number of studies**
**dating back quite a long time (e.g. Quiroga et al., 2002), but also very recent studies**
**(Siegmund et al., 2016).**

**Significance can be established quite simply with coincidence metrics, see e.g. Donges et**
**al. (2016) for an overview. In our paper we use a permutation-based scheme to find the**
**range of VACc-rates one would obtain in random data (e.g. Fig. 2). We have made all these**
**points more clear in the revised manuscript.**

Pg 6, line 20: This is a reasonable use of a supplementary figure because it provides more depth
to the analysis, but is not directly necessary for understanding the paper. It could be improved
by providing a better description. For example, "the model evaluation as shown in Fig X is
presented for a 90th percentile threshold in Fig S2 and shows . . ." If this figure is completely different from any other figure presented in the text, then you need to include a more
descriptive discussion about it in the text.

**Thanks for the suggestion! We have taken it up.**

Pg 8, lines 4-5: A figure is not a reference, don't use it like one. There needs to be discussion
about the figure and what it shows. Parts a and b need to be explained more. I am unsure as to
what the different shapes represent and the colors.

**We appreciate this point and make sure that every figure is appropriately explained and**
**discussed, both in the text and in the caption. Shapes represent the different ET datasets**
**used (diagnostic, land surface model-based, and reanalyses) - and the violins indicate the**
**distribution over all CMIP5 models (orange or dark red) and observations-based datasets**
**(gray).**

Pg 8, line 10: Fig. 3b does not say Pearson correlation, make sure it is consistent with the other
plots.

**Thanks, fixed.**

Pg 8, line 25: It is ok to reference a figure like this if you have already discussed it but since this
is the first time that fig 4 has been mentioned you need to describe what is being plotted. Also,
this is an inappropriate use of supplementary material. There needs to be explanation in the
text. How does it differ from what is being plotted in fig 4a-b. Seems like the authors are using
supplementary figures instead of actually discussing the important aspects of the analysis.

**Thanks for insisting on a proper discussion of all material. We have expanded the description and**
**discussion of this Figure, and also explained that the Supplementary Figures highlight additional**
**details in that they illustrate all individual occurrences of VACc and VACb events.**

Pg 8, line 31: I think Fig. 4 is the best and most impactful figure in this paper. Make sure that you
emphasize its importance.

**Thanks, we will emphasize the points that are raised by Fig. 4 more in the revised**
**manuscript.**

Pg 9, line 2: What does "substantially" mean? Is it significant statistically speaking?

**Thanks for this hint. We have indeed tested whether the reductions in TXx and in the 90th**
**percentile TXx (ensemble average across the 90th percentile TXx in each model) are**

**statistically significant, using a non-parametric permutation t-test. Indeed, as would be**

**expected, these reductions are significant in coupling-sensitive transitional regions (CEU,**

**CNA, partly in the Amazon); whereas in regions that are not sensitive to land-atmosphere**

**coupling the constraint does not induce any significant changes:**

[Figure]

Absolute reduction in txx−mean [°C] (warm seas., 1981−2010)

[Figure]

Absolute reduction in txx−90th perc. [°C] (warm seas., 1981−2010)

Pg 9, line 25: Again there is no information about what is plotted in the figure and the text makes
it sound relevant for understanding and interpreting the results for this paper and therefore it
should be included in the actual paper and not as supplementary material.

**The intention was to indicate that an appropriate representation of land-atmosphere**
**coupling is not only relevant for T and ET - but for related to biases in other variables as**
**well (but which is not the main focus of this paper). We will make this clear.**

Pg 10, Line 3: Again, supplementary plots are not a reference. There needs to be an explanation
of the figure.

**We have removed the reference to Fig. S13 here.**

Pg 10, Line 8: If you reference the same supplementary figure more than once, then that is a
good indication that it should be included it in the text. There is more discussion in the text on
S13 then on Fig. 5.

**Thanks for these useful comments. We will provide an extended explanation and**
**discussion in the revised manuscript.**

The manuscript by Sippel et al. addresses the reduction of ensemble temperature projections by using best estimates of soil moisture-temperature coupling diagnostics under current climate conditions. Although the technique itself has been applied in several other studies, the current application is novel and the results are highly relevant for our understanding of projections of temperature extremes. The manuscript is generally well-written and results are presented in a concise manner. The work seems technically sound, and I could not detect any major flaws in the reasoning and/or analysis, although some minor points were identified that will need to be addressed. Therefore, I believe the manuscript can be accepted for publication after minor revisions.

**We appreciate the positive evaluation of our study and research.**

I have the following remarks/observations:

• VAC is based on the 30/70th percentile, whereas the authors consider the 90th
percentile of TXx. Please motivate if and why this is justified and consistent (coupling
might be different for highest percentiles).

**Yes, indeed - The reviewer is correct in that coupling might be very different far in the tail**
**of e.g. the temperature distribution (e.g. for the highest percentiles temperature**
**extremes vs. warm, but not extremely warm conditions). This is an important caveat of**
**our study (since we are unable to address very rare events because observations-based**
**datasets are generally short in time and in many cases only available on monthly**
**resolution).**

**In the end, both choices are somewhat subjective: The choice for the 30/70th percentile**
**for determining the coupling metric has been discussed (only) briefly in the manuscript:**
**Here, the point is that the threshold choice is basically a trade-off between having enough**
**data while still looking at warm conditions (for both VACb and VACc). An additional**
**analysis using the 10th/90th percentile for computing VACb and VACc yields very similar**
**results (Figure S5), therefore increasing the confidence in our results independent of the**
**specific threshold choice, but unavoidably throws away more data.**

**For TXx, we look at both ensemble mean TXx and the 90th percentile TXx across the**
**ensemble (cf. Fig. 4d for TXx ensemble mean and Fig. 4f for TXx 90th percentile). While**

**the metric "ensemble mean TXx" is quite natural, the consideration to choose the "90th**
**percentile TXx" arose mainly from considering the "upper end" of projected TXx values**
**(similar metrics based on ensemble spread are also taken as the uncertainty bounds for**
**heat extremes, see e.g. Seneviratne et al., 2016, *Nature*). Again, changes in ensemble mean**
**TXx and 90th percentile TXx are consistent - i.e. the changes induced by the constraint**
**have the same sign, but are more pronounced for the 90th percentile of TXx. Therefore,**
**we believe that these choices are well-justifiable, and we make these considerations more**
**clear in the revised manuscript.**

**However, the inherent subjectivity of these choices also means that there is scope for**
**additional research that would look at coupling characteristics under very strong**
**heatwaves e.g. in a small number of models with a large number of ensemble members to**
**test the within-model variability in land-atmosphere coupling and its relation to extreme**
**events. We have included this point in the Discussion in Section 3.1.**

• While the manuscript has a balanced number of display items, I found the link between
the information displayed and that discussed in the text weak. Many sub-panels are
never mentioned or discussed, and too much is left for the reader to interpret. Please
make sure all relevant information in the figures is referred to, as well as all figures and
sub-panels themselves. In particular a more in depth- discussion of the results in Figures
5 and 6 is needed.

**Thanks for these suggestions. We have restructured the discussion section and put more**
**emphasis on the discussion of each single display item (please see also similar comments**
**made by Reviewer #1). In particular Fig. 5 and Fig. 6 are discussed in significantly more**
**detail (Section 3.2) and feature expanded captions. Also, in the revised manuscript we**
**refer to the individual sub-panels of the figures to make the connection between the**
**discussion and the relevant figure sub-panels clear.**

• The selection of references doesn't always to justice to work that other groups have been
doing in this area or on this specific topic. In the introduction on weighing models in
large ensembles (Page 3, lines 8–12), some examples are provide but interestingly the
ones most relevant to the current work are not cited (i.e. Fischer et al., 2012 and
Stegehuis et al., 2013). In this way, the suggestion is made that this study is the first to
apply model selection on temperature extremes. Please include references to these
works.

**Thanks for pointing this out. Our intention was by no means to claim that there has been**
**no application of other model constraints on soil-moisture temperature coupling (as the**
**Reviewer correctly points out, e.g. H in the Stegehuis et al., 2013, paper; Interannual**
**temperature variability in the Fischer et al., 2012, paper). We have discussed and cited**
**both papers mentioned in the discussion section of our manuscript, but it is true that we**
**should have referred to them also in the motivation. In the revised version we have fixed**
**this.**

Also, model selection/weighing has been applied to other aspects/fields such as snow albedo
feedback (Hall and Qu, 2006) and hydrological drought projection (Van Huijgevoort el al., 2014).

**Thanks for these references. They are indeed highly relevant to the study and we refer to**
**them both in the motivation section in the revised manuscript.**

When discussing the vegetation-atmosphere coupling index (VAC), the authors refer to previous
work from the group (e.g. Seneviratne et al., 2006; Lorenz et al., 2012) from which VAC was
developed, but not to other alternative indices that are based on a similar philosophy (for
instance the metric developed by Miralles et al., 2012, although this paper is cited in a different
context).

**Thanks for this suggestion and the reference to the Miralles et al, 2012 paper. We have**
**also fixed this.**

**References**

[revised manuscript text omitted]